# A Hierarchical Network with Fault Tolerance by a Multi-Factor Method for Neighborhood Area Network in Smart Grid

**DOI:** 10.3390/s22166218

**Published:** 2022-08-19

**Authors:** Jiatao Du, Xiaohui Li, Jie He

**Affiliations:** 1School of Information Science and Engineering, Wuhan University of Science and Technology, Wuhan 430081, China; 2Engineering Research Center for Metallurgical Automation and Measurement Technology of Ministry of Education, Wuhan University of Science and Technology, Wuhan 430081, China

**Keywords:** smart grid, neighborhood area network, multi-factor method, hierarchical network, AMI, fault tolerance

## Abstract

The neighborhood area network of a smart grid usually has hierarchical wireless communication. Due to forwarding and processing more data, the upper-layer nodes are more likely to suffer congestion and energy exhaustion. This phenomenon leads to the failure of uploading data to the control center. To solve this problem, this paper proposes a scheme for constructing a multi-factor fault-tolerant hierarchical network. This scheme firstly defines a criterion for the generation of redundant links by multi-factor method in a hierarchical wireless network with the characteristics of the neighborhood area network. Then the redundant links are used to reconstruct the existing topology of the neighborhood area network for improving fault tolerance. Finally, a greedy routing algorithm is put forward to select a proper data transmission path by bypassing low energy nodes, further reducing the failure of uploading data to the control center. The simulation results show that the proposed scheme can effectively improve the fault tolerance of the network topology of the wireless neighborhood area network and balance the network energy consumption. Compared with the original scheme, the proposed scheme improves the fault tolerance by 35% and the relative transmission rate by 21%.

## 1. Introduction

The infrastructure of the Smart Grid Communication Network (SGCN) includes Wide Area Network (WAN), Neighborhood Area Network (NAN), and custom networks such as Home Area Network (HAN), Industrial Area Network (IAN), and Business Area Network (BAN), as Figure 1 shows. NAN is the communication bridge between WAN and custom networks, and it is also the key communication infrastructure of Advanced Metering Infrastructure (AMI) [1,2] and Automated Demand Response (ADR). AMI is a complete network and system used to measure, collect, analyze and use electricity information of a user, mainly composed of smart meters and a Data Concentrator Unit (DCU). In the AMI, the DCU acts as a gateway between the Meter Data Acquisition System (MDAS) and the smart meters, which sends data to the DCU and receives MDAS commands through the DCU. Smart meters, DCU, and MDAS components together form a NAN communication network with two-way communication capabilities. AMI is divided into wired and wireless; wired AMI was widely used in the past due to its low cost, using existing power lines for communication (PLC). However, since such cables were designed to transmit power, instead of data, they are usually a harsh communication medium, suffering from frequency fading, variation of the properties of the propagation medium caused by the continuous connection, and disconnection of different loads, Electro Magnetic Interference (EMI) [3,4,5,6]; this results in unsatisfactory data transfer. Therefore, wired AMI faces great challenges. In recent years, with the continuous increase in smart devices in NAN, wireless smart meters have been widely used in NAN due to their self-organization, low cost, and rapid deployment; wireless AMI is gradually replacing wired AMI in some scenarios. Therefore, the reliability of the wireless NAN network plays an important role in the effectiveness of two-way communication and ADR of the power grid [7,8,9].

In NAN, AMI has been rapidly developed. AMI has many functions; smart meter reading is an important traditional function for AMI. For smart meter reading, smart meters generally aggregate data and transmit a few times a day. With the development of the smart grid, the power system is changing to a power-electronics-enabled distribution system, especially with the increasing penetration of distributed energy resources (DERs). To monitor and manage the electronic devices (smart inverters) and DERs (solar photovoltaic) at the grid edge, AMI with two-way communications presents great potential [10,11]. Recently, some papers and reports have investigated the monitoring and management functions of AMI, such as voltage monitoring, grid edge situational awareness, real-time demand response, and so on. In order to support the monitoring and management function of AMI, smart meters in AMI are required to report meter readings (not only the measurements for kW or kWh, but also reactive power and voltage readings) in 15 min, 30 min, or 1 h time intervals [12,13]. A large amount of metering data depends on NAN transmission. In addition, ADR commands that ensure the real-time balance between power supply and power demand also depend on NAN transmission. In some application scenarios (e.g., water metering, gas metering, and some special environments), smart meters are powered by rechargeable batteries. Even though there are many schemes to enable nodes to obtain energy from the environment, it is inevitably affected by the environment. Transmitting and processing of those data need to consume the energy of battery-powered nodes in wireless NAN. Fast exhaustion of battery energy causes node failures, even network paralysis [14]. The resulting unreliable data transmission may cause mass power grid outages and economic losses [15]. Therefore, reliable wireless network communication is the focus of NAN in a smart grid. One of the feasible methods to achieve reliable wireless communication is fault-tolerant topology control [16,17]. The fault tolerance and invulnerability of wireless NAN can be improved by appropriate topology design in the case of partial node failure [18].

Common methods for fault-tolerant topology control of wireless networks mainly include power control [19,20], adding redundant nodes [21,22], and topology evolving [23,24,25]. Those methods enhance the fault tolerance of wireless networks in different ways. However, smart devices in wireless NAN are usually mass-produced with non-adjustable power [26,27]. Adding redundant nodes will bring additional economic overhead [28,29]. Topology evolving requires wireless network topology to evolve from scratch, but the actual wireless NAN topology already exists [30]. Therefore, to optimize existing wireless AMI topology as easily and cheaply as possible, the above methods are not suitable for the fault-topology control of the wireless NAN. It is necessary to improve the fault tolerance and survivability of wireless NAN based on the characteristics of the existing wireless NAN in a smart grid.

To solve the above problems, this paper proposes a method to construct an improved hierarchical network with fault tolerance by the multi-factor method (IHN). The contributions of the paper include: (1) a criterion is defined to generate the redundant links by multi-factor method for a hierarchical wireless NAN; (2) a topology reconstruction method based on the redundant link criterion is proposed (this works as long as a network topology exists); and (3) a greedy routing algorithm is put forward to select a proper data transmission path on the reconstructed network topology to bypass low energy nodes, further reducing the failure of uploading data to the control center.

The paper is organized as follows. Notations used in this paper are summarized in Table 1. Section 2 reviews the related work and motivates this research. Section 3 explains the characteristics of the hierarchical network model for wireless NAN in a smart grid. Section 4 proposes a scheme for constructing a multi-factor fault-tolerant hierarchical network. In Section 5, a simulation analysis is carried out to prove that the proposed scheme can effectively improve the fault tolerance and energy balance of the network topology. Section 6 concludes the paper.

## 2. Related Work and Motivations

This section mainly reviews common methods that improve the fault tolerance and survivability of wireless networks. Then it identifies the technical gaps to motivate this research in this paper.

### 2.1. Related Works of Topology Control

Existing studies have shown that proper topology control can enhance the self-healing ability and survivability of NAN [31,32]. Popular methods for topology control include power control, redundancy design, and topology evolving. Those methods can improve the fault tolerance and survivability of wireless networks in different ways.

Power control is mainly to control the communication range of a node by adjusting the transmitting power. If some nodes fail, the communication reliability can be ensured by adjusting the bigger transmission range. A fault-tolerant topology control algorithm based on multicast trees is proposed, it adjusts the transmission range of each node according to the multicast tree structure, and periodically uses topology reconstruction to maintain communication [33]. A method about adaptively adjusting the communication range of nodes by the remaining energy is proposed to improve the fault tolerance of the wireless network [34]. A modified LEACH algorithm based on transmission power and node degree which can adapt to the changing of the dynamic network is proposed to discover neighbors and form into the cluster, the node can adaptively change its transmission range to ensure the network connectivity [35]. The above methods have made great progress in improving the fault tolerance of the topology. However, when using power control, it usually needs complex control strategies, resulting in the complexity of radio frequency communication.

Redundancy design mainly ensures the reliability of communication by adding redundant nodes or redundant links. It can generate redundant transmission paths as alternate routes in case of node failures. In [36], the researcher uses the maximum spanning tree to construct a robust topology, and then adds redundant links according to the remaining energy and node degree of the node to extend the life of the network and improve the stability of the network. A relay node selection algorithm is proposed to choose the proper relay node from redundant nodes for high-level connections [37]. Redundancy design can effectively tolerate node failures, but adding redundant nodes inevitably increases the cost of network deployment. If the redundant nodes such as super nodes fail, it may cause a cascading failure, so it is not suitable for wireless NAN in a smart grid.

Topology evolving is widely used in the research of fault-tolerant topology control [38]. In the literature [39], a scale-free fault-tolerant topology evolution model is proposed based on the BA model. It considers the residual energy and node degree to greatly enhance the fault tolerance of topology. In the literature [40], a scale-free network model is built in a local area, where the probability of the preferred connection depends on the remaining energy and communication traffic. It enhances the fault tolerance of the network and reduces the probability of cascading failures. The wireless topology generated by topology evolving guarantees the reliability and survivability of the wireless network to some extent. However, topology evolution requires the re-establishment of network models, which will cause a great economic burden; it is more economical to optimize the existing wireless AMI topology.

### 2.2. Technical Gaps and Motivations

Although previous research works were devoted to solving the problems that may occur in wireless networks such as balancing the network energy and tolerating node failures, those studies have obvious limitations when applied to wireless NAN. For example, wireless transmitters with adjustable power usually require complex control strategies, which usually means longer processing time and higher economic cost; they are not suitable for real-time two-way communication in smart grid NANs [41,42,43]; thus, power control is difficult to put into practice. Adding redundant nodes will increase the cost of deployment for NAN [44]. Topology evolving is only suitable for newly built wireless NAN. The existing wireless NAN topology has obvious hierarchical characteristics; a systematic scheme is required to improve fault tolerance and energy balance of the existing hierarchical wireless NAN without adding redundant nodes and installing wireless transmitters with adjustable power.

To solve the above problems, this paper mainly starts from the perspective of introducing redundant links instead of redundant nodes to improve the hierarchical wireless NAN so as not to introduce excessive network costs. The redundant link is expected within the existing communication range of the smart device so that there is no need to install a long-distance connection transmitter on the smart device in NAN.

## 3. Hierarchical Network Model of NAN

Generally, WMNs (wireless mesh networks) such as IEEE 802.15.4 g [45] are commonly considered to build NANs due to their low cost and flexibility (the network model using a cellular or a base station is a private network in the application of NANs for a smart grid; thus, the infrastructure for base station construction is time-consuming and high budget; therefore, these methods are not suitable for NANs). The network model of an AMI WMN in a smart grid can logically be treated as a tree network [46]. It is shown in the left part of Figure 2. The root node of the tree network refers to the control center and its children nodes refer to concentrators. Each concentrator has a number of children nodes (meter nodes) and each meter node may has its children nodes (other meter nodes); this relation may further extend. Moreover, each link is a wireless link; it has an obvious hierarchical structure. Data traffic in AMI usually converges to the upper nodes; the upper nodes close to the control center needs to forward and receive more data. They are more likely to suffer congestion and energy exhaustion, which causes that the underlying smart devices cannot upload the collected data to the control center. Therefore, the basic criteria for adding redundant links are:(1)Establish redundant links between the nodes at the bottom layer. Since the data are aggregated to the upper layer, the links between the nodes at the bottom layer help to share the data traffic of the nodes at the upper layer.(2)Establish redundant links between nodes at the same layer. Nodes at the same layer are usually within their existing communication ranges, and there is no need to adjust the transmit power of wireless signals when establishing redundant links. In addition, when nodes at the upper layer fail due to energy exhaustion, the links between nodes at the same lower layer can ensure that the underlying nodes can communicate with the control center through other branches. Of course, the transmission link is only logical and its characteristics are affected by many physical factors. In this paper, we assume that the physical layer constructions are the same.

Most of the nodes in wireless NAN are smart meters and other smart devices that have been running for many years. The locations of those devices are relatively fixed. Therefore, NAN is usually a relatively static wireless network. According to the characteristics of the NANs, the left part of Figure 2 can be mapped to the right part of Figure 2. As shown in the right part of Figure 2, the top of the wireless NAN usually represents the control center, node, v∈V represents the smart device, the solid lines represent the existing wireless links between smart devices, and the dashed lines represent the new wireless links. The initial network topology is Tree(V,E), and redundant links r are added to Tree(V,E) to obtain the new topology Tree(V,E+R). Therefore, the following reasonable assumptions can be made for the wireless NAN in a smart grid.

(1)Each node in a NAN network can receive and forward information; that is, each node in the Tree(V,E) can be used as a routing node.(2)Wireless signal transmission for each node with a transmission radius of *r* in NAN is isotropic; that is, the communication radius of the wireless nodes in Tree(V,E) is the same.(3)Each node in a Tree(V,E) NAN network can build a redundant link with any node in its transmission range; that is, each node in the Tree(V,E) can establish a new edge with any node in its transmission range.(4)If there is a direct link between any two nodes in a NAN network, the link will not be established repeatedly; that is, if there is already an edge between the two nodes in the Tree(V,E), no repeated edge will be constructed.(5)Each node in NAN has a limited link with other nodes, and the maximum number of links is λ; that is, the number of edges linked to each node in Tree(V,E) is at most λ.

## 4. Construction for a Multi-Factor Fault-Tolerant Hierarchical Network

Based on the basic criteria for adding redundant links mentioned above, this section firstly defines a criterion for the generation of redundant links by a multi-factor method in a hierarchical wireless NAN. Then, the redundant links are used to reconstruct the existing topology of the wireless NAN through IHN. Finally, a greedy routing algorithm is put forward to select a proper data transmission path from the redundant links. Table 1 shows the notation and symbols used in this paper.

### 4.1. The Criterion of Generation for Redundant Link

Appropriately adding a small number of redundant links can greatly enhance the fault tolerance of a wireless NAN. Adding links to a node means more data processing for the node, which will cause the energy of the node to be depleted faster. To improve fault tolerance and survivability of NAN as well as balance the energy consumption, the generation probability p(i,j) for a redundant link is defined as:(1)$$p(i,j)=e^{\left[-(\alpha +\beta +\gamma +\delta )\right]}$$
(2)$$\alpha =k_{1}\cdot D_{\mathrm{ij}}^{\mathrm{parents}}$$
(3)$$\beta =k_{2}\cdot \mu_{\mathrm{ij}}$$
(4)$$\gamma =k_{3}\cdot d_{\mathrm{ij}}$$
(5)$$\delta =k_{4}\cdot s_{\mathrm{ij}}$$
where p(i,j)∈[0,1], and α, β, γ, δ, respectively, represent four factors that affect the probability of establishing redundant links. $k_{1},k_{2},k_{3}$, and $k_{4}$ are the weight coefficients of the four factors, $k_{1}+k_{2}+k_{3}+k_{4}=1$. The size of the weight coefficients affects the priority of establishing connections between wireless nodes; that is, the larger a certain weighting coefficient is, the greater the influence of its corresponding parameter on establishing connections. The choice of the weight coefficients depends on the user’s preference. Coefficients $k_{1}$,$k_{2}$,$k_{3}$, and $k_{4}$ are all set to 0.25 in our simulation; this means that these four factors ($D_{\mathrm{ij}}^{\mathrm{parents}},\mu_{\mathrm{ij}},d_{\mathrm{ij}}$ and $S_{\mathrm{ij}}$) have the same influence on p. $D_{\mathrm{ij}}^{\mathrm{parents}},\mu_{\mathrm{ij}},d_{\mathrm{ij}}$, and $S_{\mathrm{ij}}$ are the depth of the lowest common ancestor, information rate, organizational distance, and the sum of node degree of node i and node j, respectively. These four parameters are defined in detail as follows:

$D_{\mathrm{ij}}^{\mathrm{parents}}$ is the depth of the lowest common ancestor: the lowest common ancestor means the common parent node closest to nodes i and j. The depth of node i means the minimum number of hops with which node i transfers data to the control center. $\mu_{\mathrm{ij}}$ is the sum of information rate of nodes i and j. It indicates the amount of data transmitted by the node per unit time. The node closer to the control center usually needs to transmit more data per unit time, which means its information rate is higher. $S_{\mathrm{ij}}$ is the sum of node degree of nodes i and j. The node degree means the number of nodes connected to node i. Wireless nodes can only establish connections with a limited number of nodes. $d_{\mathrm{ij}}$ is the organization distance between nodes i and j; it is defined as:(6)$$d_{\mathrm{ij}}=\sqrt{{\left(D_{i}-D_{\mathrm{ij}}^{\mathrm{parents}}\right)}^{2}+{\left(D_{j}-D_{\mathrm{ij}}^{\mathrm{parents}}\right)}^{2}-2},\;\mathrm{for}\;{}^{}D_{i}+D_{j}-2D_{\mathrm{ij}}^{\mathrm{parents}}\geq 2$$

The smaller value of $d_{\mathrm{ij}}$ can encourage links between nodes at the same layer to achieve a better energy balance as much as possible. For a given $D_{i}+D_{j}$, $d_{\mathrm{ij}}$ is minimized when $D_{i}=D_{j}$. That is to say, if the nodes are located in the same layer, the organization distance of these nodes is the smallest value [47]. $D_{i}+D_{j}-2D_{\mathrm{ij}}^{\mathrm{parents}}\geq 2$ is to avoid self-connection and repeated connections.

(1)p(i,j) increases as $s_{\mathrm{ij}}$ decreases. A smaller $s_{\mathrm{ij}}$ means fewer original links that nodes i and j have; thus they have a higher possibility to build a redundant link.(2)p(i,j) increases as the information rate $\mu_{\mathrm{ij}}$ decreases. The upper-layer node receives data traffic uploaded from its descendant nodes, so its information rate is often much higher than that of the descendant nodes. A lower information rate means the node locates at the lower layer. Since redundant links had better be established between the nodes at the bottom layer, the lower information rate between nodes i and j should have a higher possibility to build a redundant link.(3)p(i,j) increases with decreases in $D_{\mathrm{ij}}^{\mathrm{parents}}$. A smaller depth of the common ancestor node of nodes i and j means the common ancestor is closer to the control center node. Establishing a link between two such nodes means that communication is guaranteed even if the middle layer parent nodes are attacked; it can increase resilience to nodes/link failures.(4)p(i,j) increases as $d_{\mathrm{ij}}$ decreases. Links between nodes at the same layer should be encouraged to achieve a better energy balance as much as possible.

In an actual application, this method first performs a depth-first traversal of the entire network to obtain the characteristic attributes of each node in the network (such as node degree, node depth, the lowest common ancestor of two nodes), and calculates the organizational distance of each two points, and then according to the Algorithm 1, we calculate the connection probability, normalize it, and establish connections in turn according to the value of the connection probability.

### 4.2. Topology Reconstruction

Topology reconstruction in IHN mainly involves three stages:

(1)Initialization of wireless NAN. Coefficients $k_{1}$,$k_{2}$,$k_{3}$, and $k_{4}$ are all set to 0.25. This means that $s_{\mathrm{ij}}$, $\mu_{\mathrm{ij}}$, $D_{\mathrm{ij}}^{\mathrm{parents}}$, and $d_{\mathrm{ij}}$ have the same effect on p(i,j). These four coefficients can be changed with the application.(2)Computation of the generation probability for the redundant link between nodes. According to Formula (1), the generation probability for the redundant link between any node i and j that can be connected is calculated.(3)Creation of the redundant links. The maximum number of redundant links in a wireless NAN is m. The maximum number of links that can be joined by a node is λ. The generation probability of redundant links is sorted at first. Then the redundant links are established according to the probability from high to low under the constraints of m and λ.

The algorithm of topology reconstruction for a wireless NAN (Algorithm 1) is described as follows:
**Algorithm****1:** Topology reconstruction**Input**: the initial topology and
$k_{1},\;k_{2},\;k_{3},\;k_{4},\;m,\;\lambda$**Output**: the improved topology/* for each node pair
(i,j), calculate their probability of building redundant links and build links from the largest to the smallest */**for**i = 1: n** for**j = 1: n according to (1), calculate redundant link probability and store it in
p(i,j);
** end****end**Achieve the index of each node pair with
p(i,j) descending order and store them in array R;k = 1;**while** m≥0** if**$s_{R(k).i}\leq \lambda$$\;\mathrm{and}\;s_{R(k).j}\leq \lambda$ Build links between
R(k).i and R(k).j; m=m−1; k=k+1;** end****end**

### 4.3. Routing Algorithm

When a wireless NAN is reconstructed by the IHN scheme, some nodes will have multiple links that can forward data traffic due to redundant links. Thus, selecting a proper route for data transmission also has an important effect on avoiding node failures and improving the survivability of a wireless NAN. Greedy forwarding is used on the reconstructed topology to find the route from the source smart device to the control center [48]. A selection indicator function must be defined to describe which candidate device has the smallest forwarding cost to determine the best forwarding smart device. This greedy forwarding criterion can be described as a selection function, which determines the best forwarding device. Assuming that the node $v_{M}$ has M direct (single-hop) neighbor nodes $v_{1},v_{2},\ldots ,v_{M}$, and the destination node is $v_{D}$, the selection function is:(7)$$f=\min(\cos t_{1},\cos t_{2},\ldots ,\cos t_{M},)$$
where $\cos t_{i}$ refers to the cost of the node $v_{X}$ to the ith neighbor node of the destination node $v_{D}$,i∈[1,M]; this paper defines cost as follows:(8)$$\cos t_{i}=\eta D_{i}+(1-\eta )e_{i}$$
where, η is an adjustable parameter, $D_{i}$ is the depth of the ith node, and $e_{i}$ is the ith energy consumed by the first node. The parameter η determines the weight of $D_{i}$ and $e_{i}$. The routing algorithm (Algorithm 2) is as follows:
**Algorithm 2****:** Routing algorithm**Input**: the received packet (
P)**Output:** the next hop (
nexthop)/* for each neighbor of
$v_{X},$ calculate its cost */**for** each neighbord
$n_{i}$ of
$v_{X}$
$\cos t_{i}=\eta D_{i}+(1-\eta )e_{i}$**end**/* among the neighbors of
$v_{X}$, choose the minimum cost as the next hop */$\cos t_{\min}=\cos t_{i}$**for** each neighbor’s
$\cos t_{i}$ of $v_{X}$ Do** if**$\cos t_{\min}>\cos t_{i}$ then$\begin{array}{l} \cos t_{\min}=\cos t_{i};\\ \mathrm{nexthop}=v_{i}; \end{array}$** end****end**/*update the routing table of
$v_{X}$*/flagexist=0;**while**flagexist=0 and not end of routing table of
$v_{X}$ do/*
$r_{i}$ is the ith routing entry in the routing table of $v_{X}$*/** if**$r_{i}.\mathrm{destination}==P.\mathrm{destination}$**then**flagexist==1;$r_{i}.\mathrm{nexthop}=\mathrm{nexthop};$** end****end****if**flagexist==0**then**/* there is no routing entry for this packet P, need to appendnew rout information */ADD a new routing entry
$r_{\mathrm{new}};$
$r_{\mathrm{new}}.\mathrm{destination}=P.\mathrm{destination};$$r_{\mathrm{new}}.\mathrm{nexthop}=\mathrm{nexthop};$**end****if**P.destination==nexthop thendirectly send to the destination**end**

## 5. Simulation Analysis

In this section, we developed the IHN using Matlab. Our goal in conducting this simulation was to determine the advantages of fault tolerance and survivability for IHN by comparative analysis. In the simulation model, a star topology is used to collect wireless transformer sensor node (WTSN) data to a local aggregator (LA). LAs also consist of similar Zigbee transmission devices and can work in router mode. Each LA will aggregate data from all WTSNs with a scheduled time frame and following specific data structure. Then LAs send collected data to a zonal head aggregator (ZHA). A Zigbee coordinator receiver mode is set in ZHA. The transmission range of each node is 1 km. Here, star topology is followed again to aggregate all WTSN cluster data before sending it to the central server. Finally, the ZHA sends data to the central server from star topology to tree topology. We generate the initial network topology in a range of 5 km × 5 km, and the transmission range of each node is 1 km. The simulation is divided into two parts. (1) The fault-tolerance of IHN is analyzed by comparing the improved topology generated by IHN and the Original Topology (OT). (2) The survivability of IHN is analyzed by comparing the improved topology generated by IHN with the network topology generated by a long-range link (LRL). Adding a long-range link is a common method for topology improvement, usually establishing the long-range links between the control center and the node with the largest betweenness centrality, which can effectively improve the fault tolerance and survivability of the network. We first conduct simulation in a small-scale network, and then conduct simulation in a large-scale network.

### 5.1. Simulation Scenarios

According to the goals of the simulation, we design two simulation scenarios.


**Scenario 1. Verifying the fault tolerance of IHN in small-scale networks.**


To better describe the fault tolerance of a wireless NAN, we define two performance measurements: fault-tolerance C and network efficiency E. With the comparison of IHN and OT on performance measurement C and E, we analyze the advantages of fault tolerance for IHN.

In a wireless NAN network, reliable communication means that all nodes can communicate directly or indirectly with the control center. To test the fault tolerance of the network, some nodes closer to the control center (depth≤3) are randomly removed. The node degree of those nodes is usually high, and their failure will cause the breakdown of a network. Fault tolerance is determined by the ratio of the number of nodes maintaining communication with the control center to the total number of nodes. After deleting $N_{r}$ nodes in a network with a total number of N nodes, the number of nodes still communicating with the control center is normalized; S is the number of nodes maintaining communication with the control center. When some concentrator nodes are attacked, C = 1 means that all smart meter nodes can still maintain communication with the control center. It proves that the network has strong fault tolerance. Fault tolerance C is defined as follows:(9)$$C=S/(N-N_{r})$$

Network efficiency is mainly used to measure the connectivity between nodes and the overall efficiency of a network [49]. In the wireless NAN, it can be used to describe the transmission efficiency of the entire network after the network is attacked if the node cannot upload data to the control center, its shortest path to the control center will be set to infinity, and the network efficiency of the network will be reduced, which means that the connectivity of the network will also be reduced. The formula is defined as follows:(10)$$E=\frac{2}{N(N-1)}\sum_{i\neq j}\frac{1}{\mathrm{distance}(v_{i},v_{j})}$$
where E is the network efficiency (0<E<1). The larger the value of E, the higher the network efficiency. When E = 1, the network is fully connected. $\mathrm{distance}(v_{i},v_{j})$ is the shortest path length between nodes i and j in the wireless NAN. When there is no path between nodes i and j, $\mathrm{distance}(v_{i},v_{j})$ tends to infinity. Other parameters in the simulation scenario are shown in Table 2.


**Scenario 2. Verifying the survivability of IHN in small-scale networks.**


The control center in the wireless NAN network of the smart grid is usually main-powered. Energy consumption of the control center does not affect the survivability of a wireless NAN network. Other nodes in the network use battery power. Thus, the survivability of a wireless NAN depends on the balance of energy consumption among the battery-powered nodes. Other parameters in the simulation scenario are shown in Table 2.

Each node consumes energy when forwarding data, assuming that the initial energy of a node is 2 J and the energy consumption is 0.01 J when a node forwards a data packet.

Most of the data packets in a wireless NAN are transmitted from the bottom layer to the control center through several intermediate nodes. Each time a bottom node is randomly selected to start data transmission and the same routing algorithm is used on OT, IHN, and LRL to send data packets to the control center. We analyze two cases.

Case 1. The nodes at the bottom layer continuously transmit data packets until the first failure node appears in the network. This means there is a node that runs out of battery power and the network is facing a breakdown. In that case, we compare the transmission rate $N_{t}$ of LRL and IHN relative to OT.

Case 2. When transmitting the same number of data packets in LRL, IHN, and OT, we compare the distribution of residual energy of three networks. The relative transmission rates of LRL and IHN in case 1 are described as follows:(11)$$N_{\mathrm{LRL}}^{\prime }=\frac{N_{\mathrm{LRL}}}{N_{\mathrm{OT}}}$$
(12)$$N_{\mathrm{IHN}}^{\prime }=\frac{N_{\mathrm{IHN}}}{N_{\mathrm{OT}}}$$
where $N_{\mathrm{OT}}$,$N_{\mathrm{LRL}}$, and $N_{\mathrm{IHN}}$ represent the number of data packets transmitted by OT, LRL, and IHN, respectively, when the first failure node occurs in the network.


**Scenario 3. Verifying the fault tolerance and survivability of IHN in a large-scale wireless NAN network with 800 nodes.**


Wireless AMI usually has a large number of wireless smart meters. To verify the fault tolerance and survivability of IHN, large-scale simulation is carried out to evaluate the above performance indexes.

### 5.2. Simulation Results


**Simulation results of scenario 1.**


The trend of the curve for fault tolerance C of IHN is shown by the square marked line in Figure 3.

The fault tolerance of OT (OT-C) is shown by a horizontal dotted line. It can be seen that the fault tolerance of IHN (IHN-C) increases significantly with the increase in m. Compared with OT, IHN has higher fault tolerance due to the redundant links between the underlying nodes. As the number of redundant links increases, the growth trend of C slows down. This is because when the links between the underlying nodes gradually reach full connections, C will no longer be improved.

The trend of the curve for network efficiency E of IHN (IHN-E) is shown by the circle-marked line in Figure 3. The network efficiency E of OT (OT-E) is shown by a horizontal dash line. It can be seen that the network efficiency of IHN increases significantly with the increase in m. When the network has node failures, OT has very low network efficiency while IHN has higher network efficiency due to the redundant links. When 10 connections are added, the network efficiency of IHN is about 0.38, while the network efficiency of the original topology is about 0.28. Accordingly, the network efficiency is improved by about 35% using IHN.


**Simulation results of scenario 2.**


Case 1. The nodes at the bottom layer continuously transmit data packets until the first failure node appears in the network.

Figure 4 shows the distribution of residual energy when the first failure node appears for OT, IHN, and LRL. On each box in the figure, the center mark represents the median of the energy distribution, and the bottom and top sides of each box, respectively, represent the distribution of residual energy for the 25% and 75% nodes in the network. The dashed line will extend to the farthest datum that is not an outlier, and the outlier is drawn separately with the “+“ symbol. Compared with OT and LRL, the median energy value of IHN is lower, and its box is shorter. This means that the residual energy distribution of IHN nodes is more concentrated and uniform. The residual energy of most of the nodes in IHN is located between 1.38 and 1.76. The span of residual energy in LRL is the largest, and some nodes consume too much energy, while the energy of some nodes is almost not used. This is because LRL frequently uses long-range links to transmit data, and the energy consumption of nodes that add long-range links will increase very quickly. The spans of residual energy of OT and IHN are both smaller than LRL, but the energy consumption of a small number of nodes in OT is too high, such as the outlier nodes. The distribution of residual energy of outlier nodes in IHN is more even than that of OT.

Table 3 also shows the packet transmission rate of IHN and LRL network relative to OT when the first failure node appears.

The relative transmission rate of IHN is higher than that of LRL. Because the distribution of residual energy in the IHN network is more balanced than OT and LRL, the lifetime of IHN is longer than that of OT and LRL. This means IHN can transmit more data packets than OT and LRL.

Case 2. The distribution of the residual energy when transmitting the same number of data packets in LRL, IHN, and OT.

Figure 5 shows the distribution of residual energy when transmitting the same number of data packets in LRL, IHN, and OT. Compared with LRL and OT, it can be seen that the median value of IHN energy is lower and the box is shorter. This means that most of the nodes in IHN are distributed in energy from 1 to 1.7, and the distribution of residual energy in IHN is more concentrated and uniform. The box of IHN is lower than that of OT and LRL and the outlier of IHN has more residual energy than OT and LRL. This means that more nodes in IHN consume energy because they participate in data transmission. Therefore, when the same number of data packets are transmitted, the residual energy of the IHN network is more balanced.


**Simulation results of scenario**
**3.**


Figure 6 shows the distribution of residual energy when the first failure node appears for OT, IHN, and LRL in large-scale networks. We simulate a large-scale tree-like network with 800 nodes and five branches, which corresponds to a situation where a DCU receives data from multiple smart meters in wireless AMI. Compared with OT and LRL, even in large-scale networks, the median energy of IHN is lower and the box is shorter, which means that the residual energy distribution of most nodes of IHN is more concentrated and uniform; this will allow the energy of each node to be used more fully.

Figure 7 shows the distribution of residual energy when transmitting the same number of data packets in LRL, IHN, and OT on a large-scale network. Compared with LRL and OT, it can be seen that the energy distribution of LRL is similar to that of IHN. However, due to the dependence of LRL on long-range connections, the nodes with long-range connections consume energy rapidly and run out of energy. However, the remaining energy of IHN is sufficient, which also shows the superiority of IHN in energy balance.

## 6. Conclusions

To ensure the reliability of the communication in the hierarchical wireless NAN of a smart grid, this paper focuses on the construction of a hierarchical network with fault tolerance by the multi-factor method for wireless NANs in a smart grid. The proposed scheme generates redundant links for wireless NANs based on the existing hierarchical network topology without adding redundant nodes and installing power-control transmitters. Simulation results show that the proposed scheme has better fault tolerance and survivability than the original topology and the topology with long-range links, when using the same greedy routing algorithm.

Our future work will aim to improve and extend IHN. There are many directions for future research. First, we will further study the optimal value for the number of redundant links and the weight adjustment of the four factors that affect the establishment of redundant links. Second, we will implement the testing on the actual platform of wireless NANs in a smart grid. Third, we will consider the impact of physical layer factors on a smart grid NAN communication network.

## Figures and Tables

**Figure 1 sensors-22-06218-f001:**
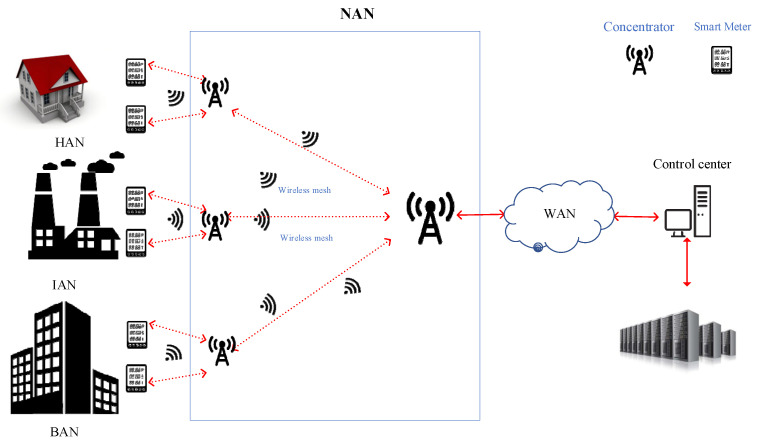
The infrastructure of smart grid.

**Figure 2 sensors-22-06218-f002:**
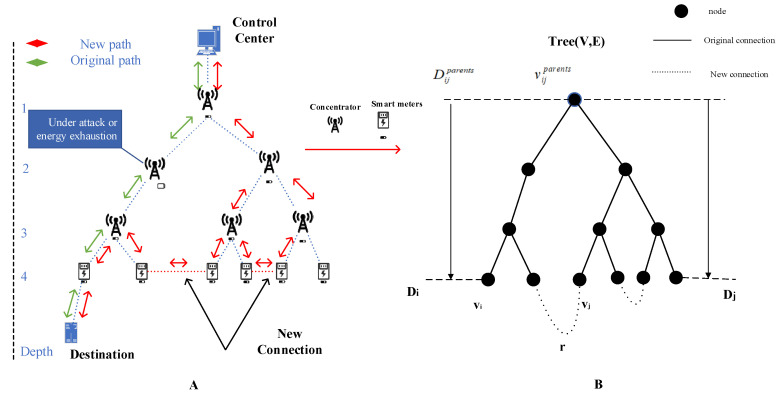
The network model of smart grid wireless NAN. (**A**) Wireless NANs; (**B**) Equivalent topology of wireless NANs.

**Figure 3 sensors-22-06218-f003:**
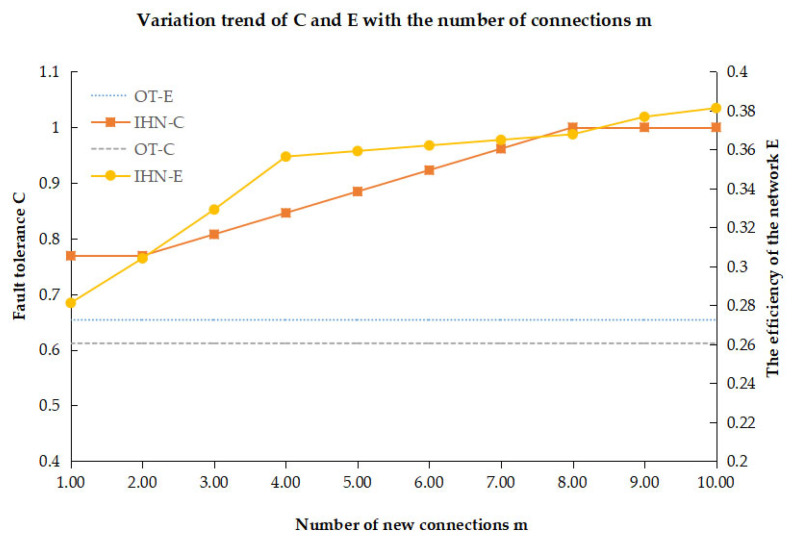
Fault tolerance and network efficiency changes with m in IHN.

**Figure 4 sensors-22-06218-f004:**
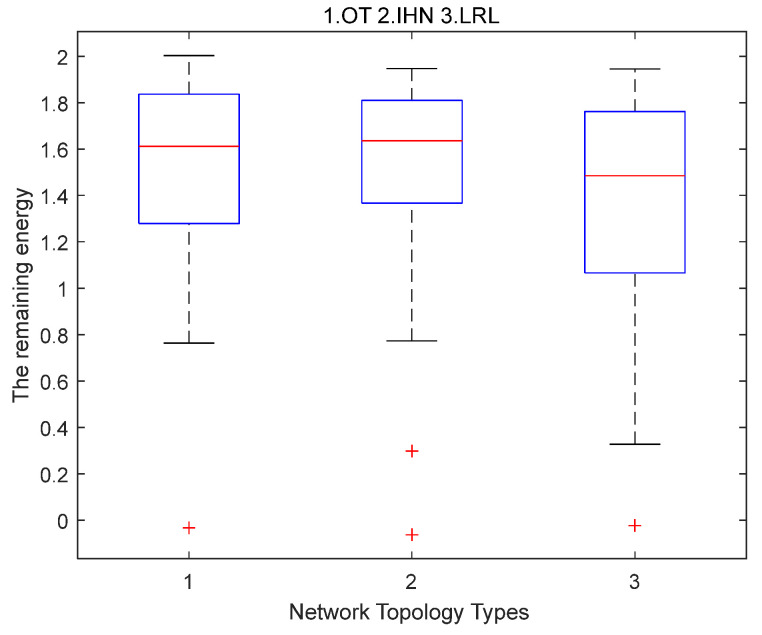
The distribution of residual energy when the first failure node appears for OT (1), IHN (2), and LRL (3) in small-scale networks.

**Figure 5 sensors-22-06218-f005:**
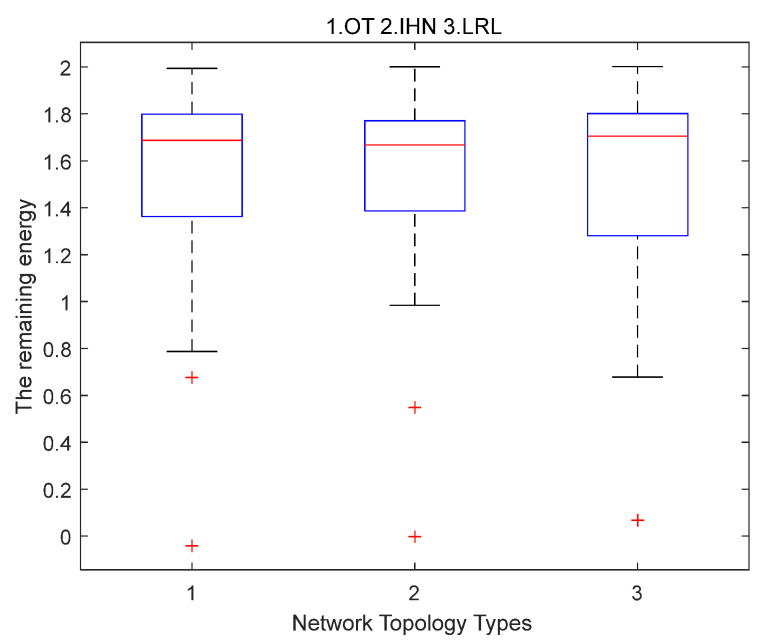
The distribution of the residual energy when transmitting the same number of data pack- ets in OT (1), IHN (2), and LRL (3) in small-scale networks.

**Figure 6 sensors-22-06218-f006:**
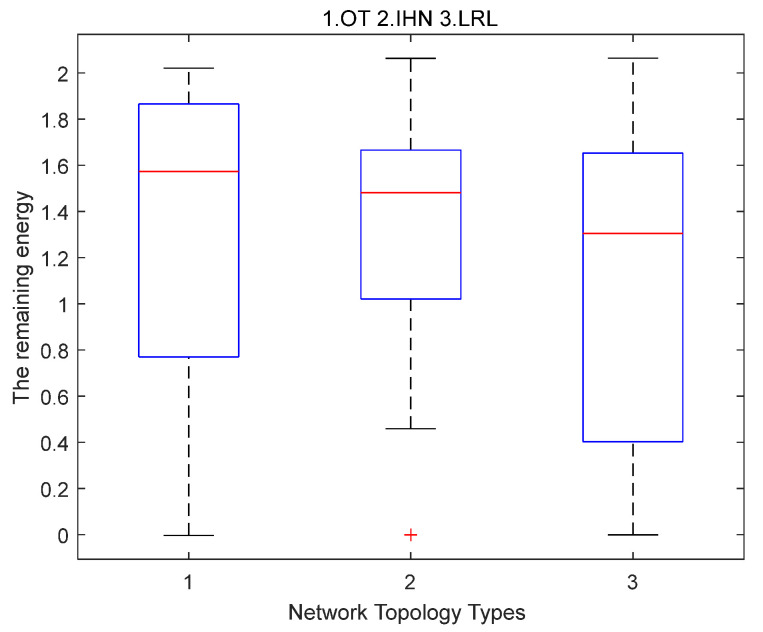
The distribution of residual energy when the first failure node appears for OT (1), IHN (2), and LRL (3) in large-scale networks.

**Figure 7 sensors-22-06218-f007:**
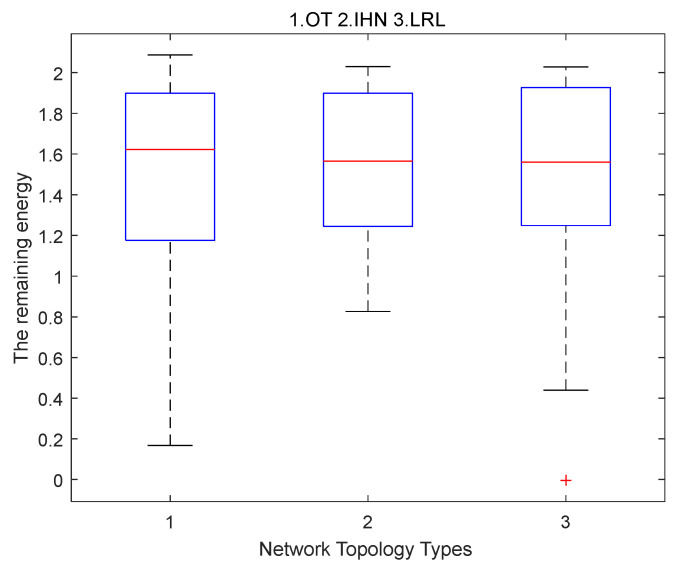
The distribution of the residual energy when transmitting the same number of data packets in OT (1), IHN (2), and LRL (3) in large-scale networks.

**Table 1 sensors-22-06218-t001:** Notations and symbols.

Parameters	Description	Parameters	Description
$D_{i},D_{j}$	The depth of node i and node j	$s_{\mathrm{ij}}$	The sum of node degree of node i and node j
$D_{\mathrm{ij}}^{\mathrm{parents}}$	The depth of the lowest common ancestor for node i and node j	p(i,j)	The probability of establishing a link between node i and node j
$v_{\mathrm{ij}}^{\mathrm{parents}}$	The lowest common ancestor node for node i and node j	V	Node set
$\mu_{\mathrm{ij}}$	The information rate of node i and node j	E	Set of initial edges in a tree
$d_{\mathrm{ij}}$	The organization distance between node i and node j	R	Set of redundant edges
$k_{1},k_{2},k_{3},k_{4}$	Self-set coefficient	n	Total number of nodes
m	Maximum number of redundant links in a wireless NAN	Tree(V,E)	The initial tree topology for a wireless NAN
λ	Maximum number of links that can be joined by a node	Tree(V,E+R)	The reconstructed tree topology with redundant edges
$s_{i}$	Node degree of node i		

**Table 2 sensors-22-06218-t002:** Simulation parameters in scenarios 1 and 2.

Parameters	Value
N	30
$N_{r}$	[0,4]
m	[1,10]
Transmission range of nodes	1 km

**Table 3 sensors-22-06218-t003:** Relative Transmission Rate for IHN and LRL When the First Node Dies.

Network	Relative Transmission Rate
LRL	1.1
IHN	1.21

## Data Availability

Data available on request due to privacy. The data presented in this study are available on request from the corresponding author.

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
