# Peer review of "A Hierarchical Network with Fault Tolerance by a Multi-Factor Method for Neighborhood Area Network in Smart Grid"

_sensors, 2022, doi:10.3390/s22166218_

Round 1

Reviewer 1 Report

My main objection to this meritorious paper is why not to make OT identical to IHN from the beginning or, at least, as soon as calculations have been performed; it would be the optimum starting point to cope with a faulty node.

Another concern: it is not clear when Algorithm 1 is called; it seems to be as soon as a node -or a link?- breaks down, but I have not seen it explicitly.

I don't understand "... are commonly considered to build NANs due to their low cost and flexibility. The network model using a cellular or a base station is ... time-consuming and high budget" (page 5, lines 170-173).

I see (3) paragraph in page 3 (260-262 lines) contradictory.

Ideas in page 2, lines 74-89, are repeated in 2.1 section. One of them, again in lines 157-158.

What does "those electronic devices (smart inverters)" refer to? It seems to be something previous.

"Nlrl, Nihn and Not" are not clearly defined in (11) and (12).

Table 1 is too far.

Some typos (line):

use user (36)

intervals[ (60)

gird (151)

is (234) (are)

root line in (6)

chose (Algorithm 2)

verifying (371)

links, when (465) (comma)

Many starting clauses have been wrongly finished with period, instead of comma. For example: "In the wireless NAN. It can be ..." (337, but not the only one).

Author Response

Please see the attachment. See reviewer 1.

Reviewer 2 Report

The manuscript addresses an interesting and important subject within the scope of the development of smart grids. 

This reviewer has the following questions/suggestions: 

a)  The authors should write some words about the interest, advantages and disadvantages of using wireless communication in smart grids. Why not wired communications as, for instance, the power line carrier technology?

b) following the recommendation in a), this reviewer believes that the authors should improve the literature review, namely by considering other options for communication in smart grids such as, for instance: 

Lars Torsten Berger, Andreas Schwager, J. Joaquín Escudero-Garzás, "Power Line Communications for Smart Grid Applications", Journal of Electrical and Computer Engineering, vol. 2013.

Melike Yigit, V. Cagri Gungor, Gurkan Tuna, Maria Rangoussi, Etimad Fadel, Power line communication technologies for smart grid applications: A review of advances and challenges, Computer Networks, Volume 70, 2014, Pages 366-383.

Carcangiu S, Fanni A, Montisci A. Optimization of a Power Line Communication System to Manage Electric Vehicle Charging Stations in a Smart Grid. Energies. 2019; 12(9):1767. https://doi.org/10.3390/en12091767

G. López et al., "The Role of Power Line Communications in the Smart Grid Revisited: Applications, Challenges, and Research Initiatives," in IEEE Access, vol. 7, pp. 117346-117368, 2019, doi: 10.1109/ACCESS.2019.2928391.

c) On page 4, the authors refer that "For example, there are few wireless transmitters with adjustable power installed on the existing smart device in NAN [37-39], thus power control is difficult to put into practice". Why is this so relevant? If the better solution is this one, the transmitters with adjustable power will appear. I believe this is not such a relevant issue! Please, explain why you think that this is a practical difficulty. 

d) The authors should better explain factors k1, k2, k3 and k4, as well as Dij, uij, Sij, explaining how they may obtain those factors in an actual application. 

e) I'm afraid I did not understand expression (6). For instance, if Di=Dj. The authors should clarify why they use this expression (why is this a good option?) and prove that the dij is minimized for Di=Dj. 

f) The authors must support expressions (8) and (10). 

Author Response

Please see the attachment. See reviewer 2

Round 2

Reviewer 1 Report

I see it ready to be published.

Reviewer 2 Report

I want to thank the reviewers for their answers. 

I believe the manuscript is suitable to be accepted.

However, I recommend text editing. 

This manuscript is a resubmission of an earlier submission. The following is a list of the peer review reports and author responses from that submission.

Round 1

Reviewer 1 Report

The authors proposed a scheme for constructing a multi-factor hybrid fault-tolerant hierarchical network. They presented  simulation results  to show that their proposed scheme can effectively improve the fault tolerance of the network topology of the wireless neighborhood area network and balance the network energy consumption. The paper is well organized, but there are numerous issue need to be addressed as follows:

Abstract: you need to write the percentage of improvement that your proposed scheme is achieved. 

Section 3: Table 1. Notations and Symbols. Please decrease the line spacing

The authors mentioned that on page 6, line 199 “ four factors that affect the probability of establishing redundant” what is the definition of those parameters?

The algorithms should be numbered for clarity.

The word “Where”” after each equation should be in lowercase.

Section 4: is there any reference for OT??

I suggest to combine Table 2 and 3 in one table.

Simulation results of scenario 1 is not well discussed.

In Figure 3, the legend should be included all curves and horizontal lines, not just mentioned then in the test. This figure is not acceptable.

Figures 4,5,6,7 are not very clear, please enhance their qualities. 

What is the overall performance percentage of your proposed scheme comparing with the original one?
